# Higher Adiponectin Levels in Children and Adolescents with T1D Probably Contribute to the Osteopenic Phenotype through the RANKL/OPG System Activation

**Charalampos Tsentidis** [1,*], **Dimitrios Gourgiotis** [2], **Lydia Kossiva** [1], **Antonios Marmarinos** [2], **Artemis Doulgeraki** [3] and **Kyriaki Karavanaki** [1]

1   Diabetes Clinic, 2nd Department of Pediatrics, Athens University Medical School, "P&A Kyriakou" Children's Hospital, 11527 Athens, Greece; lydiakossiva@hotmail.com (L.K.); kkarav@yahoo.gr (K.K.)
2   Laboratory of Clinical Biochemistry—Molecular Diagnostics, 2nd Department of Pediatrics, Athens University Medical School, "P&A Kyriakou" Children's Hospital, 11527 Athens, Greece; dgourg@med.uoa.gr (D.G.); antonios_marmarinos@hotmail.com (A.M.)
3   Department of Bone and Mineral Metabolism, Institute of Child Health, "Aghia Sophia" Children's Hospital, 11527 Athens, Greece; doulgeraki@yahoo.com
*   Correspondence: tsentidis@yahoo.gr

**Abstract:** Background: Diabetes mellitus is an increasing global health emergency, with serious complications (including osteoporosis). Leptin and adiponectin are among the least-investigated possible contributing factors of T1D low bone mass. Methods: In this case-control cross-sectional analysis, we evaluated 40 pairs of T1D children and adolescents and controls. We evaluated body diameters and skinfolds, leptin, adiponectin, lipids and lipoproteins, bone metabolic markers and DXA parameters of BMD and fat percentage. Results: Leptin levels were comparable between groups and correlated well with body mass parameters. Adiponectin levels were found to be higher in the patient group and correlated with higher levels of HbA1c, triglycerides and s-RANKL. Conclusions: In this study, leptin levels were no different, but adiponectin levels were found to be higher in children and adolescents with T1D and correlated with diabetic metabolic derangement indices and s-RANKL in the patient group. Adiponectin can be considered a surrogate marker of T1D in young patients' metabolic status and probably contributes to the diabetic low bone mass phenotype via activation of the RANKL/OPG metabolic pathway.

**Keywords:** adiponectin; leptin; type 1 diabetes; children; adolescents; bone metabolism; osteopenia; osteoporosis; fat mass; lipids



## 1. Introduction

Diabetes mellitus is an increasing global health emergency with serious complications, affecting about half a billion individuals worldwide. There is an increasing incidence for both type 1 (T1D) and type 2 (T2D) diabetes in children and adolescents, and the number of new cases of T1D in ages 1–19 years is estimated to be 149,500 patients per year [1]. Among the detrimental complications of the disease is osteoporosis, even in the lower ages. Children and youth with T1D were found to have lower bone mineral content (BMC), lower areal bone density (aBMD) and deficits in trabecular bone density and micro-architecture [2]. These deficits, along with microvascular complications, are probably responsible for the observed increased fracture risk across the life span of T1D patients [3]. An increasing proportion of the medical literature focuses on the pathophysiology of lower bone mass accrual in young T1D patients, and the interaction between bone and adipose tissue is one of the systems being lately investigated.

A crosstalk between bone and adipose tissue is well-established [4–6], with peptides such as the lipokines superfamily acting as mediators of this link, balancing between

energy conservation and bone quality [4]. Leptin and adiponectin were the first pleiotropic adipokines that were studied regarding their bone morphogenic role. Leptin, a 16 kDa protein, mainly secreted by adipose tissue, plays an important role in the energy regulation of various systems [7,8]. A growing body of cell culture, animal and human studies has revealed conflicting results about the effect of leptin on the skeleton, leading to a theory of dual leptin action: one peripheral direct anabolic effect on osteoblasts and chondrocytes and one central indirect catabolic effect through sympathetic neural activity. The net action seems to be species specific and analogous to homeostatic body conditions [5,6,9]. Adiponectin is a 28 kDa protein, excreted mainly by adipose tissue. Generally, it has an inverse relationship with obesity and insulin resistance and decreases circulating lipid levels [10]. Although most animal studies have been inconclusive about its action on bone, some cellular line studies have shown an osteoanabolic effect of adiponectin on osteoblasts [5,11] and inhibition of osteoclasts [5,11], but others have documented an inhibition of osteoblasts [12–14], especially via the RANKL/OPG (receptor activator of nuclear factor-KappaB ligand/osteoprotegerin) system [15–17].

In this context of elucidating the role of leptin and adiponectin in bone metabolism, we performed the study of these lipokines in young patients with T1D and controls in an effort to investigate their associations with clinical and biochemical parameters and thus hypothesize their possible role in developing the low bone mass phenotype of T1D patients.

## 2. Materials and Methods

### 2.1. Study Population

The protocol of this case-control cross-sectional study has been extensively described in previous publications [18,19]. In brief, we evaluated 40 consecutive Greek Caucasian children and adolescents with T1DM and 40 healthy age- and gender-matched controls during a three-year period at the Diabetic Outpatient Clinic of 2nd Department of Pediatrics, University of Athens. Controls were patients' friends, classmates or relatives and completely healthy. Inclusion criteria for patients were age greater than 5 years and disease duration of more than 2 years, while controls had to be over the age of 5 years and be perfectly healthy. Exclusion criteria for both patients and controls were the use of vitamin and mineral supplements or dietary restrictions, the coexistence of other chronic diseases (with a possible impact on bone metabolism) and the use of medications affecting bone turnover, like corticosteroids and antipsychotic or neuroleptic drugs, as well as adolescents on contraceptives or subjects having sustained a fracture three months preceding the study.

### 2.2. Clinical Evaluation

After a thorough clinical examination during the routine visit, body weight and standing height were recorded and BMI Z-scores (SDS-BMI) were evaluated from national normative data. The pubertal stage of participants was evaluated by a single examiner (Ch.T.) using the Marshall and Tanner visual scale. Apart from body diameters (waist, hip, branchial, thigh and calf), typical skinfolds were evaluated using the Harpenden skinfold caliper (dial graduation: 0.20 mm; measuring range: 0 to 80 mm; measuring pressure: 10 gms/mm$^2$; constant over range, accuracy: 99.0%; repeatability: 0.20 mm) in order to evaluate the subcutaneous fat distribution. The mean value of two repeated measurements was used in the final analysis.

### 2.3. Biochemical Assays

After an overnight fast, morning blood samples (08:00–10:00 a.m.) were obtained and stored at −80 °C until final biochemical evaluation.

Leptin was measured in plasma with the enzyme immunoassay (EIA) method using a human leptin kit provided by Biovendor (Biovendor, 62100 Brno, Czech Republic) with a detection limit of 0.2 ng/mL (intra-assay CV 4.2%, inter-assay CV 6.7%).

Adiponectin was measured in plasma with the enzyme immunoassay (EIA) method using a human adiponectin kit provided by Affymetrix eBioscience (Affymetrix eBioscience, Bender MedSystems GmbH, Campus Vienna Biocenter 2, 1030 Vienna, Austria) with a detection limit of 0.01 ng/mL (intra-assay CV 3.1%, inter-assay CV 4.2%).

Bone metabolism markers (Dickkopf-1, sclerostin, total soluble receptor activator of nuclear factor-KappaB ligand (s-RANKL), osteoprotegerin (OPG), osteocalcin and C-terminal telopeptide crosslinks of type I collagen (CTX)) were measured in plasma with the sandwich enzyme-linked immunosorbent assay (ELISA) method, while intact parathyroid hormone (PTH) and insulin-like growth factor 1 (IGF1) were measured in serum with the electrochemiluminescence immunoassay (ECLIA) method that was previously described [19].

Total serum alkaline phosphatase (ALP), glycosylated hemoglobin (HbA1c), lipids and lipoproteins were measured with routine biochemical assays in the biochemistry laboratory of "P.& A. Kyriakou" children's hospital.

### 2.4. Bone Density and Fat Mass Evaluation

After informed consent from all parents that was obtained in advance, total body less head (TB-BMD) and lumbar spine bone mineral density (L1-L4-BMD) were evaluated with dual energy X-ray absorptiometry (DXA) in the department of bone and mineral metabolism at the institute of child health in "Aghia Sophia" children's hospital. In addition to BMD evaluation, whole body DXA was utilized to evaluate body fat mass and percentage and body lean mass.

The ethics committee of our hospital and the university of Athens medical school approved the research protocol (protocol code: 6340/12.04.2011).

### 2.5. Statistical Analysis

All statistical analyses and data management were performed using STATA for Windows v16, (StataCorp. 2019. Stata Statistical Software: Release 16. College Station, TX, USA: StataCorp LLC). Data are expressed as mean $\pm$ SD for normally distributed variables and median (interquartile range) for skewed variables.

Power analysis indicated that a sample size of 40 pairs had 85% power to detect a mean adiponectin difference (delta) of 4500 with a significance level (alpha) of 0.05 (two-tailed). Exploratory data analysis was performed for all numerical variables both graphically and statistically, using the Shapiro–Wilk criterion. Numerical variables that were right-skewed were log-transformed, while two variables were square root-transformed to fulfill normality assumption.

After mathematical transformation of skewed numerical variables, the paired *t*-test was used for comparisons between patients and controls, whereas Fisher's exact test was used for categorical variables' comparison between the aforementioned groups. In each group, associations of leptin and adiponectin with demographic, somatometric, metabolic and bone parameters were performed with Pearson's correlation coefficient for univariate comparisons, while partial correlation coefficient was utilized for multiple comparisons (adjusted for gender, chronological age and SDS-BMI). The main research idea was to compare different association patterns between patients and controls, possibly indicating activation of certain pathophysiological pathways, contributing to osteoporosis in T1D patients. $p \leq 0.05$ was considered significant.

### 3. Results

Patients' and controls' characteristics are presented and compared in Table 1. Although patients and controls had comparable levels of SDS-BMI and DXA fat percentage, the levels of low density cholesterol (LDL-C), apolipoprotein B100 (Apo-B) and triglycerides were higher in the patient group (Table 1). These results are indicative of the metabolic derangement of diabetes.

**Table 1.** Study variable distribution in T1DM patients and controls. Data are presented as actual numbers (%) for categorical variables, mean ± SD for normally distributed variables and median (range) for skewed variables. Comparisons are between patients and controls.

| Characteristics | Overall (*n* = 80) | Controls (*n* = 40) | Patients (*n* = 40) | *p* |
|---|---|---|---|---|
| *Gender (boys/girls)* | 36/44 | 18/22 | 18/22 | 0.58 ** |
| *Chronological age (years)* | 13.02 ± 3.39 | 13.04 ± 3.53 | 12.99 ± 3.3 | 0.39 * |
| *SDS BMI* | 0.290 ± 0.93 | 0.298 ± 0.92 | 0.286 ± 0.83 | 0.52 * |
| *DEXA fat percentage (%)* | 27.6 ± 9.7 | 28.3 ± 8.8 | 26.2 ± 10.1 | 0.35 * |
| *Physical activity* $(Kcal \times kg^{-1} \times day^{-1})$ | 34.2 ± 3.9 | 35.88 ± 4.69 | 33.91 ± 3.8 | 0.15 * |
| *HbA1c (%)* | 6.50 ± 2.23 | 4.75 ± 0.18 | 8.25 ± 1.95 | <0.001 * |
| *(IFCC, mmol/mol)* | (48 ± 24.4) | (28 ± 2.0) | (67 ± 21.3) | |
| *12-month HbA1c (%)* | 6.41 ± 2.00 | 4.75 ± 1.18 | 8.06 ± 1.58 | <0.001 * |
| *(IFCC, mmol/mol)* | (47 ± 21.9) | (28 ± 2.0) | (65 ± 17.3) | |
| *Fasting glucose (mg/dL)* | 124.8 ± 41.2 | 82.23 ± 5.63 | 167.4 ± 71.3 | <0.001 * |
| *(mmol/L)* | (6.92 ± 2.28) | (4.56 ± 0.32) | (9.26 ± 3.94) | |
| *HS-CRP* | 0.32 (0.15, 0.97) | 0.39 (0.15, 1.13) | 0.32 (0.15, 0.72) | |
| *Log (HS-CRP)* | −0.86 ± 1.0 | −0.77 ± 1.05 | −0.98 ± 0.94 | 0.22 * |
| *T-CHOL (mg/dL)* | 160 ± 27 | 156 ± 25 | 164 ± 28 | 0.09 * |
| *LDL-C (mg/dL)* | 89 ± 21 | 85 ± 21 | 93 ± 21 | 0.05 * |
| *HDL-C (mg/dL)* | 61 ± 14 | 62 ± 13 | 60 ± 14 | 0.26 * |
| *Triglycerides (Tg's) (mg/dL)* | 58 ± 34 | 52 ± 15 | 65 ± 46 | 0.04 * |
| *Non-HDL-C (mg/dL)* | 97 ± 26 | 94 ± 23 | 101 ± 29 | 0.14 * |
| *Apo-B (mg/dL)* | 64 ± 15 | 61 ± 13 | 67 ± 17 | 0.03 * |
| *Apo-A1 (mg/dL)* | 147 ± 24 | 147.2 ± 24.1 | 147.5 ± 24.5 | 0.47 * |
| *Lp(a) (mg/dL)* | 9.7 (4.9, 20.8) | 9.3 (4.3, 21.9) | 10.4 (5.4, 17.4) | |
| *Log (Lp(a))* | 2.3 ± 0.9 | 2.2 ± 0.9 | 2.3 ± 0.9 | 0.35 * |
| *T-CHOL/HDL-C* | 2.7 ± 0.6 | 2.6 ± 0.5 | 2.8 ± 0.6 | 0.06 * |
| *LDL-C/HDL-C* | 1.5 ± 0.5 | 1.4 ± 0.5 | 1.6 ± 0.4 | 0.06 * |
| *LDL-C/APO-B* | 1.3 ± 0.1 | 1.39 ± 0.2 | 0.38 ± 0.1 | 0.42 * |
| *Tg's/HDL-C* | 1.06 ± 0.8 | 0.9 ± 0.4 | 1.2 ± 1.1 | 0.06 * |
| *APO-B/APO-A1* | 0.44 ± 0.12 | 0.42 ± 0.1 | 0.46 ± 0.13 | 0.06 * |
| *Non-HDL-C/HDL-C* | 1.7 ± 0.6 | 1.6 ± 0.5 | 1.8 ± 0.6 | 0.06 * |
| *Leptin (ng/mL)* | 6.6 (2.6, 14.3) | 8.2 (3.07, 13.1) | 5.1 (1.5, 15.9) | |
| $\sqrt{Leptin}$ | 2.7 ± 1.4 | 2.8 ± 1.3 | 2.5 ± 1.4 | 0.16 * |
| *Adiponectin (ng/mL)* | 13,586 (10,192, 18,667) | 12,503 (9693, 15,431) | 16,979 (11,683, 23,682) | |
| *Log (adiponectin)* | 9.5 ± 0.4 | 9.4 ± 0.4 | 9.7 ± 0.5 | 0.005 * |
| *Adiponectin/leptin* | 2380 (914, 5304) | 1628 (764, 4224) | 3404 (1074, 11,226) | |
| *Log (adiponectin/leptin)* | 7.8 ± 1.3 | 7.5 ± 1.1 | 8.1 ± 1.4 | 0.01 * |
| *Insulin dose* $U \times Kgr^{-1} \times day^{-1}$ | | 0.96 ± 0.22 | - | |

* Student's *t* test (one-sided); ** Fisher's exact test; SDS BMI—standard deviation score of body mass index; HbA1c—glycosylated hemoglobin; HS-CRP—high sensitivity C-reactive protein; T-CHOL—total cholesterol; HDL-C—high density cholesterol; LDL-C—low density cholesterol; Apo-B—apolipoprotein B100; Apo-A1—apolipoprotein A1; Lp(a)—lipoprotein α.

Adiponectin and leptin exhibited skewed distributions (adiponectin skewness 1.45, kurtosis 5.53, Figure 1; leptin skewness 1.31, kurtosis 4.72, Figure 2). Leptin levels were found to be comparable in patients and controls, but adiponectin levels were significantly higher in T1D patients. The difference in the adiponectin to leptin ratio was mainly attributed to the difference in adiponectin levels (Table 1).

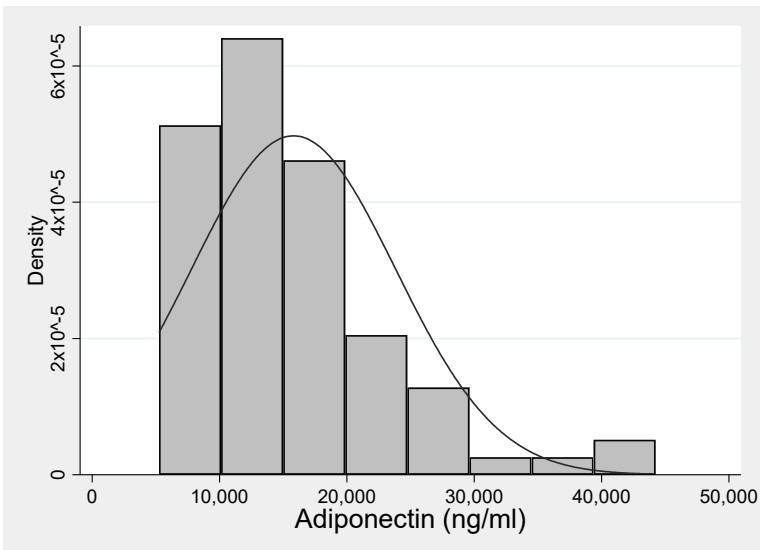

**Figure 1.** The skewed distribution of adiponectin.

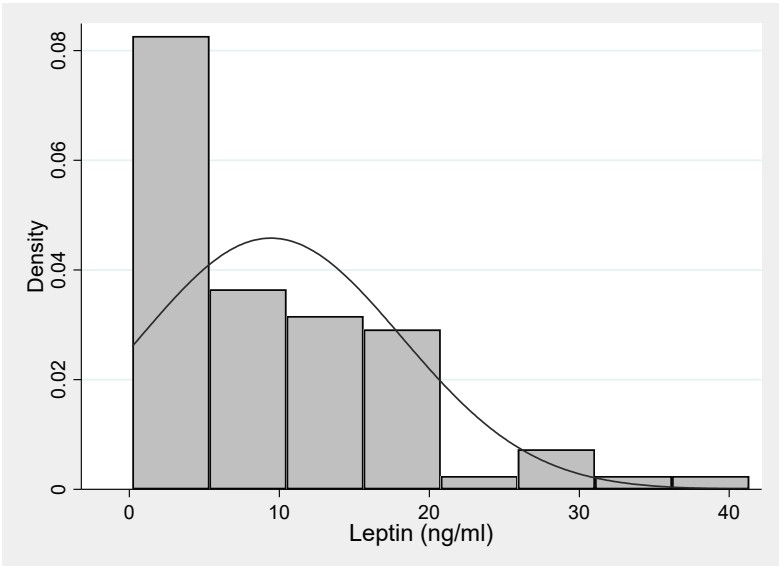

**Figure 2.** The skewed distribution of leptin.

We also evaluated the associations of adiponectin and leptin with demographic, somatometric, metabolic and bone parameters in patients and controls, as presented in Tables 2 and 3. In the patient group, leptin exhibited strong and significant associations with gender, Tanner pubertal stage, SDS-BMI, fat mass and percentage, lean mass, waist and hip diameters and almost all skinfolds, while its association with whole body BMD Z-score might reflect the general association with body mass, as already described. These associations were largely observed in the control group as well (Tables 2 and 3). Adiponectin exhibited a different pattern of associations. There were strong correlations in patients with glycosylated hemoglobin (both basal and 12-month HbA1c), triglycerides and soluble receptor activator of nuclear factor-KappaB ligand (log (s-RANKL)) that were not observed in the controls (Tables 2 and 3).

**Table 2.** Associations of leptin and adiponectin with demographic, somatometric and metabolic parameters in patients and controls, using Pearson's coefficient for univariate associations and partial correlation coefficient for multiple comparisons, adjusted for gender, age and SDS-BMI.

| Pearson's and Partial Correlation Coefficients, *p*-Value | √Leptin | | | Log (Adiponectin) | | |
|---|---|---|---|---|---|---|
| | Overall (*n* = 80) | Controls (*n* = 40) | Patients (*n* = 40) | Overall (*n* = 80) | Controls (*n* = 40) | Patients (*n* = 40) |
| *gender* * | **0.47, <0.001** | 0.26, 0.09 | **0.65, <0.001** | 0.016, 0.88 | −0.22, 0.17 | 0.21, 0.17 |
| *age* * | 0.20, 0.06 | 0.10, 0.50 | 0.29, 0.06 | **−0.25, 0.01** | **−0.34, 0.02** | −0.21, 0.18 |
| *Tanner stage* * | **0.24, 0.02** | 0.13, 0.41 | **0.36, 0.02** | −0.19, 0.07 | −0.27, 0.08 | −0.16, 0.29 |
| *SDS-BMI* * | **0.43, <0.001** | **0.34, 0.03** | **0.48, 0.001** | **−0.25, 0.02** | −0.10, 0.50 | −0.26, 0.09 |
| *DXA fat percentage* | **0.42, <0.001** | 0.20, 0.22 | **0.70, <0.001** | −0.03, 0.73 | −0.14, 0.40 | −0.004, 0.98 |
| *DXA fat mass* | **0.43, <0.001** | **0.27, 0.10** | **0.65, <0.001** | 0.11, 0.35 | 0.16, 0.32 | 0.12, 0.47 |
| *DXA lean mass* | −0.20, 0.07 | −0.01, 0.91 | **−0.38, 0.02** | −0.12, 0.27 | −0.25, 0.12 | −0.06, 0.73 |
| *physical activity* | −0.13, 0.24 | −0.15, 0.34 | −0.07, 0.68 | −0.20, 0.07 | −0.22, 0.17 | −0.05, 0.74 |
| *insulin dose* | - | - | −0.01, 0.93 | - | - | −0.09, 0.59 |
| *diabetes duration* | - | - | 0.17, 0.31 | - | - | −0.0002, 0.99 |
| *HbA1c* | −0.03, 0.74 | 0.002, 0.98 | −0.18, 0.26 | **0.39, <0.001** | 0.03, 0.98 | **0.39, 0.01** |
| *12-month HbA1c* | −0.05, 0.65 | 0.002, 0.98 | −0.22, 0.17 | **0.36, 0.001** | 0.038, 0.98 | **0.37, 0.02** |
| *waist diameter* | **0.36, 0.001** | **0.38, 0.01** | **0.40, 0.01** | 0.03, 0.76 | **0.35, 0.03** | −0.23, 0.17 |
| *hip diameter* | **0.27, 0.01** | 0.25, 0.12 | **0.34, 0.03** | −0.09, 0.43 | −0.12, 0.47 | −0.02, 0.87 |
| *branchial diameter* | −0.11, 0.31 | 0.17, 0.30 | −0.19, 0.23 | 0.05, 0.60 | 0.04, 0.78 | 0.04, 0.80 |
| *thigh diameter* | **0.22, 0.05** | 0.17, 0.29 | 0.30, 0.07 | −0.049, 0.66 | 0.17, 0.31 | −0.09, 0.56 |
| *calf diameter* | −0.04, 0.67 | 0.14, 0.40 | −0.11, 0.48 | 0.14, 0.19 | 0.06, 0.69 | 0.16, 0.33 |
| *biceps skinfold* | **0.26, 0.01** | 0.26, 0.11 | 0.22, 0.17 | −0.13, 0.24 | 0.01, 0.92 | −0.31, 0.06 |
| *triceps skinfold* | **0.38, <0.001** | **0.31, 0.05** | **0.43, 0.006** | 0.12, 0.27 | 0.03, 0.82 | 0.09, 0.56 |
| *subscapular skinfold* | **0.27, 0.01** | 0.24, 0.13 | **0.34, 0.03** | −0.10, 0.35 | −0.18, 0.27 | −0.08, 0.62 |
| *suprailiac skinfold* | **0.40, <0.001** | **0.40, 0.01** | **0.45, 0.004** | 0.04, 0.69 | 0.05, 0.75 | −0.001, 0.99 |
| *abdominal skinfold* | **0.27, 0.01** | 0.15, 0.34 | **0.36, 0.02** | 0.09, 0.41 | −0.12, 0.47 | 0.23, 0.15 |
| *thigh skinfold* | **0.40, <0.001** | **0.36, 0.02** | **0.48, 0.002** | 0.19, 0.09 | 0.17, 0.30 | 0.09, 0.57 |
| *calf skinfold* | **0.39, <0.001** | **0.55, <0.001** | 0.22, 0.18 | −0.02, 0.81 | 0.13, 0.44 | −0.11, 0.48 |
| *T-CHOL* | 0.17, 0.13 | 0.013, 0.93 | 0.24, 0.13 | 0.15, 0.17 | −0.06, 0.68 | 0.12, 0.46 |
| *triglycerides* | 0.08, 0.47 | −0.04, 0.78 | 0.14, 0.39 | **0.31, 0.005** | −0.02, 0.86 | **0.35, 0.02** |
| *HDL-C* | −0.009, 0.93 | −0.06, 0.71 | −0.13, 0.42 | 0.12, 0.26 | 0.20, 0.22 | 0.01, 0.91 |
| *LDL-C* | 0.13, 0.25 | 0.07, 0.67 | 0.18, 0.27 | 0.05, 0.61 | −0.11, 0.48 | 0.014, 0.93 |
| *Apo-B* | 0.06, 0.57 | −0.07, 0.65 | 0.16, 0.36 | 0.09, 0.45 | −0.27, 0.11 | 0.12, 0.50 |
| *Apo-A1* | −0.05, 0.68 | −0.14, 0.39 | −0.11, 0.52 | 0.08, 0.47 | 0.14, 0.39 | 0.03, 0.85 |
| *log (Lp(a))* | 0.04, 0.71 | 0.10, 0.56 | 0.008, 0.96 | 0.04, 0.73 | 0.06, 0.62 | −0.004, 0.98 |

\* Pearson's correlation coefficient; SDS BMI—standard deviation score of body mass index; HbA1c—glycosylated hemoglobin; T-CHOL—total cholesterol; HDL-C—high density cholesterol; LDL-C—low density cholesterol; Apo-B—apolipoprotein B100; Apo-A1—apolipoprotein A1; Lp(a)—lipoprotein α.

**Table 3.** Associations of leptin and adiponectin with biochemical and bone parameters in patients and controls, using partial correlation coefficient for multiple comparisons, adjusted for gender, age and SDS-BMI.

| Partial Correlation Coefficients, *p*-Value | √Leptin | | | Log (Adiponectin) | | |
|---|---|---|---|---|---|---|
| | Overall (*n* = 80) | Controls (*n* = 40) | Patients (*n* = 40) | Overall (*n* = 80) | Controls (*n* = 40) | Patients (*n* = 40) |
| *log (HS-CRP)* | 0.06, 0.61 | 0.007, 0.97 | 0.43, 0.06 | −0.15, 0.27 | −0.24, 0.19 | 0.19, 0.40 |
| *IGF-1* | 0.03, 0.78 | 0.04, 0.77 | 0.09, 0.55 | −0.16, 0.14 | −0.01, 0.94 | −0.21, 0.20 |
| *log (PTH)* | −0.05, 0.66 | −0.12, 0.44 | 0.12, 0.45 | 0.08, 0.43 | 0.04, 0.78 | 0.20, 0.21 |
| *√ALP* | −0.15, 0.17 | −0.09, 0.57 | −0.26, 0.13 | 0.01, 0.89 | 0.01, 0.92 | −0.06, 0.73 |
| *dickkopf-1* | 0.009, 0.93 | 0.24, 0.13 | −0.18, 0.27 | −0.009, 0.93 | −0.06, 0.72 | −0.07, 0.64 |
| *sclerostin* | −0.02, 0.80 | −0.20, 0.21 | −0.21, 0.20 | 0.12, 0.26 | 0.08, 0.63 | 0.07, 0.67 |
| *log (CTX)* | 0.13, 0.25 | 0.22, 0.18 | 0.06, 0.70 | −0.03, 0.76 | 0.12, 0.44 | −0.04, 0.77 |
| *log (osteocalcin)* | 0.06, 0.56 | 0.15, 0.34 | 0.01, 0.94 | −0.004, 0.96 | 0.19, 0.24 | −0.07, 0.65 |
| *log (s-RANKL)* | 0.20, 0.06 | 0.25, 0.12 | 0.20, 0.22 | 0.11, 0.30 | −0.18, 0.28 | **0.35, 0.03** |
| *OPG* | −0.20, 0.07 | −0.26, 0.11 | −0.21, 0.20 | **0.23, 0.03** | 0.16, 0.33 | 0.16, 0.32 |
| *L1-L4 BMD Z-score* | −0.15, 0.17 | −0.10, 0.27 | −0.10, 0.51 | **−0.22, 0.05** | −0.22, 0.17 | −0.17, 0.30 |
| *WB BMD Z-score* | −0.20, 0.06 | −0.09, 0.58 | **−0.33, 0.04** | −0.17, 0.12 | −0.10, 0.53 | −0.13, 0.41 |

SDS BMI—standard deviation score of body mass index; HS-CRP—high sensitivity C-reactive protein; IGF1—insulin-like growth factor 1; log (PTH)—logarithm of intact parathyroid hormone; √ALP—square root of total alkaline phosphatase; log (CTX)—logarithm of C-terminal telopeptide crosslinks of type I collagen; log (osteocalcin)—logarithm of osteocalcin; OPG—osteoprotegerin; log (s-RANKL)—logarithm of soluble receptor activator of nuclear factor-KappaB ligand; Z-score L1-L4 and total body BMD—standard deviation score of lumbar spine and total body less head bone mineral density.

Leptin associations seemed to follow the body mass distribution, especially the fat mass and percentage and also subcutaneous fat. Adiponectin associations seemed to follow the metabolic derangement of diabetes and probably were connected with the derangement of the RANKL/OPG bone metabolic system (Tables 2 and 3).

## 4. Discussion

In this work we studied leptin and adiponectin distributions in young patients with T1D and controls, as well as lipokine associations with various demographic, somatometric, metabolic and DXA factors. Leptin concentration was found to be comparable in both groups, and its associations followed the body mass distribution, as expected by its physiology. We also found that adiponectin had higher values in the patient group, and its associations followed the metabolic derangement of diabetes and also correlated with the derangement of the RANKL-OPG bone metabolic system, probably contributing to the pathogenesis of the lower bone mass of patients with T1D.

Leptin, and more extensively adiponectin, has been studied in various conditions. From a metabolic view, in patients with T2D and insulin resistance, in whom the risk of cardiovascular disease is high, adiponectin was found in lower concentrations and leptin was found in higher concentrations [10,20,21]. In T1D, which is characterized by insulin depletion, leptin is expected to be found in lower concentrations, while adiponectin concentrations should be higher.

In the published data of young patients with T1D, leptin was found either in lower [22] or comparable concentrations to the controls [23] and was related positively to body fat mass in both patients and controls [23]. Leptin was not associated with [23] or was negatively associated with HbA1c and was positively related to insulin dose [22], and showed no relation to the disease course [23]. Furthermore, leptin was negatively correlated with OC and showed no correlation with bone mineral density (BMD) or Z-score values of BMD [22]. In a similar way, in adult T1D patients, leptin did not differ from the controls [20] and was positively correlated with BMI, fat body mass and daily insulin dose [24].

In previous studies of Polish, North American and German T1D young patients, adiponectin was found in higher concentrations than in the controls [22,23,25,26] and was either not associated [22,23,26] or was positively associated with HbA1c in young patients [25]. Adiponectin serum concentrations were not related to the body fat content in the study groups and showed no relation to the disease course [23]. Adiponectin was also positively related to OPG, but was found unrelated to the bone mineral density (BMD) or Z-score values of BMD [22].

In T1D adults, adiponectin was similarly found in higher concentrations [24,27,28] and was further elevated in patients with poor metabolic control [29]. One study reported that this increase was mainly explained by an elevation in the biologically active high molecular weight (HMW) subform [27]. Adiponectin was also inversely correlated with BMI and daily insulin dose [24] and found to be uncorrelated with osteocalcin or the RANKL/OPG system [24].

The differences found in our study, compared with these historic data of young patients, could be greatly related to the different sample size of each study, regarding chronological age and diabetes duration, different proportions of patients with good metabolic control, ethnic differences, high variance in variables such as the Tanner pubertal stage and gender proportions and, finally, differences in measurement techniques.

The associations of leptin and adiponectin in this work should be viewed as metabolic signals of adipose tissue to other systems, especially bone tissue in the context of diabetes. Leptin exhibited an expected neutral profile with significant associations for fat mass indices in both groups. In contrast to leptin, and in line with previous studies of T1D children and adolescents, adiponectin in our dataset exhibited a different correlation profile with no association with fat mass or BMI. The higher values in the patient group and the positive association with HbA1c, LDL-C and triglycerides probably indicate that adiponectin is a general alerting metabolic signal.

T1D is characterized by insulin depletion and glucose starvation of muscle and fat cells [30]. We have to place adiponectin in this particular frame of pathophysiology as a signal to preserve and utilize available energy. As leptin has a duality in action, depending on the homeostasis state, so must adiponectin. So, hyperadiponectinemia should be secondary due to hyperglycemia and insulin reserve depletion, as adiponectin induces insulin sensitivity [31,32]. There are several indications that adiponectin signals cellular starvation and energy depletion. Newly diagnosed patients with T1D and detectable c-peptide had lower levels of adiponectin than patients with longer standing diabetes and undetectable c-peptide, both with HbA1c < 6.0% [28]. In adult patients with hyperglycemia in an acute emergency department setting, normalization of blood glucose caused a decrease in adiponectin levels, independent of diabetes type and/or body weight, indicating again that the restoration of energy flow in cells with insulin reduced adiponectin levels [33]. In a subgroup of patients with new onset T1D in a pediatric T1D longitudinal study, adiponectin levels increased during follow-up as native insulin secretion declined [26]. Adiponectin can thus be used as a surrogate marker of metabolic state, as higher adiponectin levels were found in adult patients with T1D and worse metabolic control [34], and as a surrogate marker of beta cell function in T1D patients, as it was inversely correlated with fasting and meal-stimulated c-peptide [35].

Adiponectin in the state of hyperglycemia and relative or absolute insulin deficiency should thus be recognized as an alerting signal to preserve and utilize available energy and probably signals the skeleton in an equal manner to reduce energy expenditure. It probably has a differential action on bone, depending on homeostasis state and other signals that are all integrated on osteocytes and osteoblasts. Adiponectin's negative action on bone mass has been previously recognized. In a large cohort of 4927 normal children, the investigators concluded that, independent of fat mass, lean mass and height, adiponectin was associated with lower bone mass in childhood, predominantly due to action on relative endosteal expansion [36].

In young patients with T1D, there is also an activation of the RANKL/OPG pathway, with higher levels of both RANKL and OPG, probably contributing to lower bone mass, as we have previously shown [18]. A negative effect of adiponectin on bone via the RANKL/OPG pathway has also been previously documented [15–17,37], while in another study it has been suggested that one of the mechanisms adiponectin influences osteoclastogenesis is by increasing osteoclast formation via stimulating RANKL and inhibiting OPG production in the osteoblasts [38]. In the present study, higher adiponectin levels were significantly and positively correlated with s-RANKL in the patient group, while, in a previous work, higher adiponectin levels were positively correlated with OPG [22].

These two observations probably indicate that, in T1D young patients characterized by hyperglycemia and insulin deficiency, higher adiponectin levels signal the bone tissue to preserve energy and reduce bone modeling via the RANKL/OPG pathway, possibly contributing to the low bone mass phenotype of T1D patients. The negative signal is probably stronger in patients with worse metabolic control, as adiponectin increases in parallel with HbA1c.

There are some additional examples of adiponectin's negative action on bone tissue in other pathologic conditions via the RANKL/OPG pathway. Adiponectin exacerbated collagen-induced arthritis via enhancing Th17 response and prompting RANKL expression [39]. Adiponectin also had a negative effect on bone metabolism in adolescent idiopathic scoliosis osteopenia via ADR1-RANKL/OPG, a RANKL/OPG pathway activated by adiponectin receptor 1 (ADR1) [40].

This study contributes to the current literature by recognizing adiponectin as a marker of cellular starvation and energy depletion and a negative signal of bone metabolism in order to reduce bone energy expenditure in favor of other critical systems for survival. Adiponectin, along with HbA1c and glucose variance, could be used in the clinical setting as an indicator of metabolic derangement and worsening glucose control. It could also indicate carbohydrate metabolism normalization in efficiently treated patients and could

be a surrogate index of other metabolic systems' normalization, such as adipose and bone tissue after effective treatment.

Although this study inferred interesting results, it has several limitations. The small number of participants in both groups limits the extent of investigating the adipokines' association with other covariates. Additionally, the cross-sectional nature of this work impedes its ability to reveal causal relationships between investigated factors. Both limitations constrain the results of this study from extrapolating to all T1D young patients. Finally, study results may not be generalizable to individuals of different ethnicities.

Further multicenter and longitudinal studies are necessary in order to clarify the role of adipokines in bone metabolism of patients with diabetes. Future research should be conducted using gene expression analyses in blood or tissue from bone of patients and controls in order to reveal RNA-Seq differential expression and thus elucidate activated pathways in diabetic bones.

## 5. Conclusions

In this case-control cross-sectional analysis, leptin concentrations exhibited no differences, but adiponectin was found in higher concentrations in children and adolescents with T1D and was correlated with diabetes metabolic derangement indices and s-RANKL in the patient group. Adiponectin can be considered a surrogate marker of T1D young patients' metabolic status and probably contributes to the diabetic low bone mass phenotype via activation of the RANKL/OPG metabolic pathway.

**Author Contributions:** Conceptualization, C.T. and K.K.; methodology, C.T. and K.K.; software, C.T.; validation, D.G., A.M. and A.D.; formal analysis, C.T.; investigation, C.T.; resources, L.K. and A.D.; data curation, C.T. and A.M.; writing—original draft preparation, C.T.; writing—review and editing, L.K., A.D., D.G. and K.K.; visualization, C.T.; supervision, K.K.; project administration, K.K. and D.G.; funding acquisition, C.T. and K.K. All authors have read and agreed to the published version of the manuscript.

**Funding:** This research was partially funded by Hellenic Association For The Study And Education Of Diabetes Mellitus providing the two kits for measuring leptin and adiponectin.

**Institutional Review Board Statement:** The study was conducted in accordance with the Declaration of Helsinki, and approved by the Institutional Review Board of "P. & A. Kyriakou" Children's Hospital (protocol code 6340/12.04.2011).

**Informed Consent Statement:** Informed consent was obtained from all subjects involved in the study.

**Data Availability Statement:** Original data sheet can be provided upon reasonable request.

**Acknowledgments:** We thank the administration of the Hellenic Association for the study and education of diabetes mellitus for the donation of leptin and adiponectin measurement kits.

**Conflicts of Interest:** The authors declare no conflict of interest. The funders had no role in the design of the study; in the collection, analyses, or interpretation of data; in the writing of the manuscript; or in the decision to publish the results.

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
