# Peer review of "Higher Adiponectin Levels in Children and Adolescents with T1D Probably Contribute to the Osteopenic Phenotype through the RANKL/OPG System Activation"

_endocrines, doi:10.3390/endocrines4040051_

Round 1
Reviewer 1 Report
Comments and Suggestions for Authors
1. Introduction: I kindly suggest making the introduction more concise. It may not be necessary to include data from animal studies. Additionally, consider shifting the findings of previous studies to the discussion section and focus more on the novelty of your study in the introduction.
2. Study Objective: Given the cross-sectional study design, it's challenging to establish the role of adipokines in developing low bone mass. Instead, consider redefining the study objective to focus on identifying associations.
3. Results Section - Table 1: In Table 1, the authors mention higher T chol and Apo-B levels in the patient group, but the p-values are not significant. Please clarify or provide context for these findings.
4. Table 2: Table 2 appears extensive and challenging to follow. I recommend either moving some data to a supplementary table or creating a separate table for adipokines and DXA results to enhance clarity.
5. Please maintain consistency in reporting p-values throughout the manuscript, either rounding to 2 or 3 digits.
6. Discussion: I recommend making the discussion more concise, not exceeding two pages, and ensuring better organization. The authors should discuss their study results and explain them in the context of recent literature findings.
7. The discussion currently extensively discusses findings from other studies, making it difficult to discern if they are pertinent to the authors' study results or findings from prior studies. Please clarify and streamline this section.
8. Lines 258-271: These lines require more clarification. Please provide additional context or explanation.
9. Limitations: It's important to acknowledge that the study results may not be generalizable to individuals of different ethnicities. Please include this limitation.
10. Given the well-established negative association of adiponectin with bone metabolism and the limited utility of using adiponectin as surrogate levels for assessing low bone density in clinical practice, please elaborate on how this study contributes to the current literature and how these findings can be applied to treat or prevent low bone density in children and adolescents.
Author Response
Dear reviewers,
Thank you for your kind and constructive remarks. They helped to improve the quality of the manuscript and to form a meaningful discussion. Below are the remarks and the respective answers. The main text was corrected with tracking changes and red font during corrections.
- Remark: Introduction: I kindly suggest making the introduction more concise. It may not be necessary to include data from animal studies. Additionally, consider shifting the findings of previous studies to the discussion section and focus more on the novelty of your study in the introduction.
Answer: Respective changes in text were made. Animal data were removed but some data from cell lines were kept because they showed the negative effect of adiponectin on bone morphometry. The descriptive text about previous studies was shifted to the discussion section
- Remark: Study Objective: Given the cross-sectional study design, it's challenging to establish the role of adipokines in developing low bone mass. Instead, consider redefining the study objective to focus on identifying associations.
Answer: Respective changes in text were made.
- Remark: Results Section - Table 1: In Table 1, the authors mention higher T chol and Apo-B levels in the patient group, but the p-values are not significant. Please clarify or provide context for these findings.
Answer: T-CHOL had indeed comparable values in groups and was corrected.
- Remark: Table 2: Table 2 appears extensive and challenging to follow. I recommend either moving some data to a supplementary table or creating a separate table for adipokines and DXA results to enhance clarity.
Answer: Table 2. was divided as suggested in Table 2. and Table 3. and the two new tables now describe the two distinct entities of general metabolism and bone metabolism. Throughout the text, we made an effort to maintain a shorter number of variables. If we excluded some data in the supplementary table we would lose descriptive information that is important for the formation of main hypotheses. So, we decided to keep the current set of data but separated the set into three new tables.
- Remark: Please maintain consistency in reporting p-values throughout the manuscript, either rounding to 2 or 3 digits.
Answer: P-values were rounded to 2 digits, except in a few cases of values <0.01
- Remark: Discussion: I recommend making the discussion more concise, not exceeding two pages, and ensuring better organization. The authors should discuss their study results and explain them in the context of recent literature findings.
Answer: The discussion section was rearranged in order to have a meaningful presentation of the main hypothesis, which is that adiponectin is 1. a marker of cell starvation and energy deprivation and 2. in this context it signals the skeleton to preserve energy via RANKL/OPG system activation. Many paragraphs were rewritten but in order to keep the evidence references of our hypothesis, which are significant previous studies, the text couldn’t be reduced much.
- Remark: The discussion currently extensively discusses findings from other studies, making it difficult to discern if they are pertinent to the authors' study results or findings from prior studies. Please clarify and streamline this section.
Answer: The paragraphs, that support evidence of our hypothesis were rewritten and simplified.
- Remark: Lines 258-271: These lines require more clarification. Please provide additional context or explanation.
Answer: Those lines were rewritten with a simplified approach.
- Remark: Limitations: It's important to acknowledge that the study results may not be generalizable to individuals of different ethnicities. Please include this limitation.
Answer: This addition was made in the limitations section.
- Remark: Given the well-established negative association of adiponectin with bone metabolism and the limited utility of using adiponectin as surrogate levels for assessing low bone density in clinical practice, please elaborate on how this study contributes to the current literature and how these findings can be applied to treat or prevent low bone density in children and adolescents.
Answer: A paragraph with the potential role of adiponectin in a clinical setting was added.
Reviewer 2 Report
Comments and Suggestions for Authors
Tsentidis and colleagues' research titled “Higher adiponectin levels in children and adolescents with T1D probably contribute to the osteopenic phenotype through the RANKL/OPG system activation. The study is intriguing, with a well-organized and structured manuscript. The results showcase a commendable comparative analysis between healthy individuals and those diagnosed with Type 1 Diabetes. The reported findings signify significant progress within the realm of Diabetes treatment research. The authors contribute systematically to the existing body of literature in this particular study area.
Minor Revision:
The author did not specify whether the study included patients with other disorders or those who had previously recovered from non-diabetes-related conditions. While the study focused on a Greek Caucasian population group and examined various parameters related to metabolic disturbances and their connection to T1D, it did not discuss whether similar metabolic disturbances are observed in other population groups as studied by various other research groups. Furthermore, it would be advantageous to acknowledge any potential limitations of the study and suggest future avenues for further research. The study contains several typographical errors, such as "grour," "12month," and more, which require correction.
Comments on the Quality of English Language
Need improvement and typographical errors correction
Author Response
Dear reviewers,
Thank you for your kind and constructive remarks. They helped to improve the quality of the manuscript and to form a meaningful discussion. Below are the remarks and the respective answers. The main text was corrected with tracking changes on and red font during corrections.
Tsentidis and colleagues' research titled “Higher adiponectin levels in children and adolescents with T1D probably contribute to the osteopenic phenotype through the RANKL/OPG system activation. The study is intriguing, with a well-organized and structured manuscript. The results showcase a commendable comparative analysis between healthy individuals and those diagnosed with Type 1 Diabetes. The reported findings signify significant progress within the realm of Diabetes treatment research. The authors contribute systematically to the existing body of literature in this particular study area.
Minor Revision:
Remark: The author did not specify whether the study included patients with other disorders or those who had previously recovered from non-diabetes-related conditions.
Answer: A respective addition clarifying the control group was made in the text.
Remark: While the study focused on a Greek Caucasian population group and examined various parameters related to metabolic disturbances and their connection to T1D, it did not discuss whether similar metabolic disturbances are observed in other population groups as studied by various other research groups.
Answer: The ethnicity of the other study groups was added to the text. Additionally, the possible bias due to ethnicity was added in the limitations section
Remark: Furthermore, it would be advantageous to acknowledge any potential limitations of the study and suggest future avenues for further research.
Answer: Respective changes were made in the text
Remark: The study contains several typographical errors, such as "grour," "12month," and more, which require correction.
Answer: There was an extensive review for typographical and syntax errors and they were corrected.